# Targeting Specific Sites in Biological Systems with Synchrotron X-Ray Microbeams for Radiobiological Studies at the Photon Factory

**Akinari Yokoya [1],\* and Noriko Usami [2]**

1. Institute for Quantum Life Science, National Institutes of Quantum and Radiological Sciences and Technology, 2-4 Shirakata, Tokai, Ibaraki 319-1106, Japan
2. Photon Factory, Institute of Materials Structure Science, High Energy Accelerator Research Organization (KEK), 1-1 Oho, Tsukuba, Ibaraki 305-0801, Japan; noriko.usami@kek.jp
\* Correspondence: yokoya.akinari@qst.go.jp; Tel.: +81-0-70-3943-3412

**Abstract:** X-ray microbeams have been used to explore radiobiological effects induced by targeting a specific site in living systems. Synchrotron radiation from the Photon Factory, Japan, with high brilliance and highly parallel directionality is a source suitable for delivering a particular beam size or shape, which can be changed according to target morphology by using a simple metal slit system (beam size from 5 μm to several millimeters). Studies have examined the non-targeted effects, called *bystander cellular responses*, which are thought to be fundamental mechanisms of low-dose or low-dose-rate effects in practical radiation risk research. Narrow microbeams several tens of micrometers or less in their size targeted both the cell nucleus and the cytoplasm. Our method combined with live-cell imaging techniques has challenged the traditional radiobiological dogma that DNA damage is the only major cause of radiation-induced genetic alterations and is gradually revealing the role of organelles, such as mitochondria, in these biological effects. Furthermore, three-dimensionally cultured cell systems have been used as microbeam targets to mimic organs. Combining the spatial fractionation of X-ray microbeams and a unique ex vivo testes organ culture technique revealed that the tissue-sparing effect was induced in response to the non-uniform radiation fields. Spatially fractionated X-ray beams may be a promising tool in clinical radiation therapy.

**Keywords:** X-ray microbeam; cell targeting; live-cell imaging radiation biology

## 1. Introduction

Cells in multicellular systems have been targeted by irradiation with ionizing radiation or UV lasers in mechanistic studies of radiobiological phenomena. These phenomena ultimately lead to carcinogenesis induced by low-dose radiation from the environment or medical applications, such as microsurgery or microbeam radiation therapy (MRT) (see review by Drexler and Ruiz-Gómez [1]). Experimental evidence has shown that cell-to-cell communications between exposed and unexposed cells surrounding the exposed cells play an important role in inducing certain kinds of genetically critical effects in the unexposed cells, known as the *bystander response* (see reviews by Prise et al. [2], Holl and Hei [3]). Nagasawa and Little [4] performed pioneering work in which they exposed 1% of Chinese hamster ovary cells in a culture dish to alpha particles and then measured the induction of sister chromatid exchanges (SCE). They found the SCE efficiency was about 30%, even though a small number of cells (1%) were traversed by an alpha particle. A similar enhancement of SCE induction by alpha-particle irradiation of a small percentage of cells was reported by Deshpande et al. [5]. These studies challenged the radiobiological dogma that radiation damage to DNA is the only major cause of genomic alteration. It was inferred that certain kinds of signal transfer from the exposed cells

to neighboring unexposed cells induce genetic instability irrespective of whether the cells experience direct radiation hit [6]. Early studies observed the bystander effect but did not elucidate the entirety of the underlying mechanisms of the bystander effect. Because the alpha particles hit the cells in a stochastically random manner, the cells that were hit could not be identified spatially and temporally in the culture dish.

As accelerator techniques have progressed, microbeams have been developed that can target particular sites in a field. Various high-LET charged particle microbeams of protons, He, carbon, or much heavier ions, such as uranium, have been used to study the correlation between an extremely localized track structure and the resulting complex DNA damage induction followed by enzymatic repair in cultured cells (see reviews by Tobias et al. [7], Barberet and Seznec [8], and Kobayashi et al. [9]). On the other hand, the synchrotron X-ray microbeam provides a uniformly exposed small area in a wide range of biological targets, from DNA to animals. This is a specific advantage of this microbeam compared with ion particle irradiation or focused laser beams. In this review, recent advances in mechanistic studies of specific sites in biological systems targeted with a parallel X-ray microbeam from a synchrotron facility, the Photon Factory, in Tsukuba, Japan, are described. In particular, the use of the X-ray microbeam combined with live-cell imaging techniques is highlighted in relation to investigating the risks of low-dose radiation exposure, as well as radiation therapy for cancer treatment.

## 2. Early Studies of Bystander Effects using X-Ray Microbeams

In early studies using X-ray microbeams in radiation biology in the 1990s, characteristic X-rays emitted from conventional X-ray sources were used. Carbon-K (278 eV), aluminum-K (1.5 keV), or titanium-K X-rays (4.5 keV) were focused by diffraction lenses and X-ray Fresnel zone plates to obtain a narrow beam with a spot size of less than 1 μm (see review by Prise et al. [10]). Soft X-rays were used to mimic low-dose exposure of higher-energy photons, such as γ-rays emitted from contaminating radioactive materials released by a nuclear power plant accident. γ-rays deposit their energy into living systems through a limited number of tracks of lower-energy electrons as a result of initial Compton scattering and the energy deposition patterns are similar to that of the photoelectric effect of soft X-rays. Exposing cultured mammalian cells to focused carbon-K X-ray microbeams revealed that targeting a single cell in a population results in a certain fraction of damaged cells (micronucleus or apoptosis induction) throughout the dish (Schettino et al. [11]). These observations indicated that a typical mechanism of bystander signal transfer is the transfer of signal molecules from the irradiated cells to neighboring cells through the medium.

## 3. Synchrotron X-Ray Microbeams in Radiation Biology

Although these early studies demonstrated some important aspects of the bystander effect, the variability of the beam size of the characteristic X-rays was limited by the focusing optics (X-ray Fresnel zone plates). Changing the beam size or beam shape to conform to the target morphology provides a powerful probe for the precise analysis of radiobiological effects (Figure 1). Kobayashi et al. [12] developed an X-ray microbeam irradiator with 5.35 keV monochromatic X-rays using synchrotron radiation from the Photon Factory, KEK, Japan. Synchrotron radiation from a low-emittance electron storage ring provides an ideal brilliant light source. The intense photon flux at the sample position and the highly parallel directionality allows the size and shape of the beam to be controlled using a simple metal slit system (from 5 μm to a few mm range). The system was also designed to irradiate cytoplasm only, by shielding the nucleus with a gold mask (Figure 1C) [12]. The method of evaluating the absorbed dose in the exposed area should also be discussed. Due to the short range of secondary electrons (1.1 μm maximum) produced by 5.35 keV X-ray exposure, the dose delivered outside of the irradiated area was negligible. Fukunaga et al. [13] used PHITS code to calculate dose profiles of various spatially fractionated microbeams with widths of 12.5, 50, and 200 μm for ex vivo testes organ culture (details are described in the section "Clinical applications of X-ray microbeams"). The estimated dose outside of the exposed area was less than 0.25% of that of the exposed one.

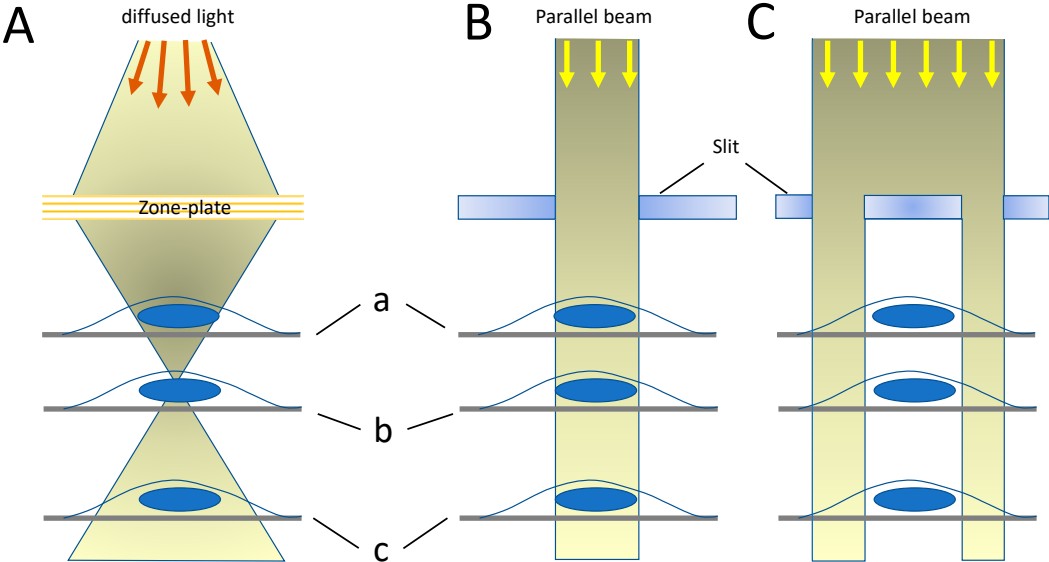

**Figure 1.** Schematic of microbeam exposure of a cell at positions labeled 'a', 'b', and 'c'. When X-rays from a conventional source are focused using a diffraction lens, such as an X-ray Fresnel zone plate, the beam size depends on the distance from the lens to cell (**A**). A simple metal slit system is used to restrict quasi-parallel synchrotron X-rays spatially to target the cell nucleus (**B**) or cytoplasm (**C**).

Using the X-ray microbeam irradiator, the bystander cell-killing effect was induced when five nuclei in normal human fibroblast cells (WI-38) were targeted with a $10 \times 10 \ \mu m^2$ beam, and the effect was suppressed by pretreatment with an inducible nitric oxide (NO) synthase inhibitor (aminoguanidine) or a NO scavenger (carboxy-PTIO). These results suggest that mainly NO, which is a free radical, mediates the effect [14]. Maeda et al. [15] also investigated the survival fraction of Chinese hamster lung cells (V79) based on microcolony formation ability after exposure of whole cells or cell nuclei to a $50 \times 50$ or $10 \times 10 \ \mu m^2$ beam, respectively, in the low-dose (<1 Gy) region. For microbeam exposure, the mammalian cells were cultured in a specially designed dish with a thin polyimide film bottom (Figure 2). They reported that, in the low-dose region, significant hyper-radiosensitivity with a minimum survival fraction of about 60% at 0.5 Gy was observed for nucleus irradiation compared with the dose-response curve obtained for whole-cell irradiation (Figure 3). They suggested that the radiation effects on cytoplasm caused by whole-cell irradiation might play a role in suppressing the hyper-radiosensitivity. These results also challenge the radiobiological dogma that DNA damage solely determines cell survival fraction in a dose-dependent manner. The cytoplasmic alterations induced by irradiation are an important modifier of radiobiological effects.

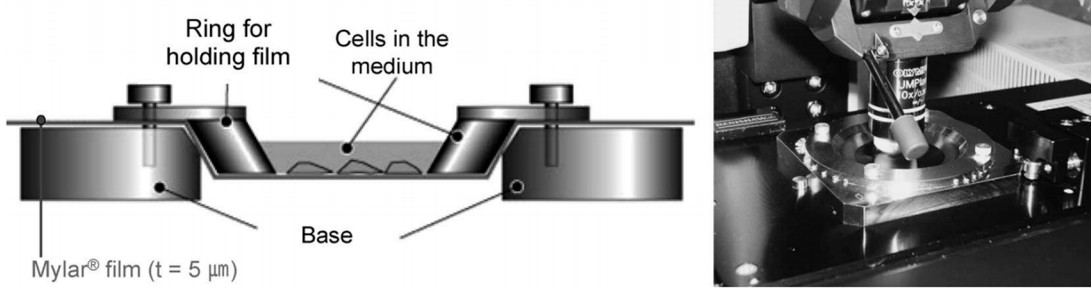

**Figure 2.** Schematic of the Mylar dish designed for microbeam irradiation (**left**) and the dish on the irradiation stage (**right**) in the Photon Factory, KEK, Tsukuba, Japan. This figure is reprinted from Maeda et al. [15] with permission.

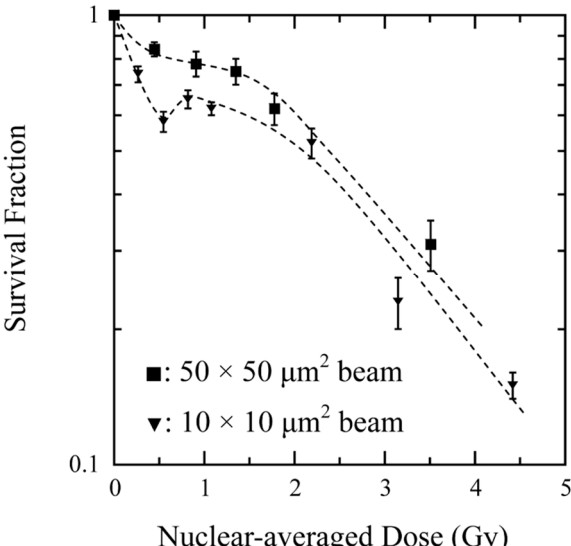

**Figure 3.** Survival curves of V79 cells irradiated with $10 \times 10$ (▼) or $50 \times 50$ μm$^2$ (■) X-ray beams. Standard errors are indicated by error bars. This figure is reprinted from Maeda et al. [15] with permission. The horizontal axis shows the nuclear-averaged dose or absorbed dose for the $10 \times 10$ or $50 \times 50$ μm$^2$ beam irradiation, respectively.

## 4. Cytoplasmic Irradiation with X-Ray Microbeams

Mitochondria, which are distributed throughout the cytoplasm, are organelles that are crucial in synthesizing the bioenergetic molecule, ATP. Although various mitochondrial effects have been found in many studies using charged particles (see review by Zhou et al. [16]), microbeams deposit their energy in limited areas along their trajectories, called *tracks*, and sometimes high-energy secondary electrons emitted from the tracks may reach cell nuclei and cause DNA damage.

To observe the mitochondrial effects separately from the nucleus damage, X-ray microbeams with short-range secondary electrons provide an ideal probe for exposing only the cytoplasm. In some experiments, the cytoplasm has been irradiated selectively using a metal mask to shield the cell nucleus from beam exposure. Using a conventional X-ray source, Ghita et al. [17] investigated DNA damage in human breast cancer cells (MDA-MB-231) and primary fibroblast cells (AG01522) with a low energy X-ray microbeam (278 eV) combined with gold nanoparticles (GNPs) as radiosensitizing agents. They targeted the cell nucleus and cytoplasm selectively, and found that, even when the cytoplasm was targeted, the number of 53BP1 foci, which is an indicator of DNA double-strand break induction, gradually increased during 3 h of incubation for the cancer cells, whereas it decreased for the normal fibroblast cells. The increase or decrease was less than 20%. The areas exposed to the microbeam in the cytoplasm were 5 μm in diameter; hence, part of or most of the mitochondria in the cells maintained their high membrane potential for ATP production. Exposing all the mitochondria to X-ray microbeams is also important. We have investigated the mitochondrial effects by using a deformational synchrotron X-ray microbeam from the Photon Factory (KEK, Japan) (Figure 4) to expose cell nuclei or cytoplasm uniformly. Cytoplasmic irradiation was performed using a $60 \times 60$ μm$^2$ beam with a 22 μm diameter gold mask to shield the cell nucleus. The particular mitochondrial active sites with high membrane potential were visualized by fluorescent chemical probes. Unexpectedly, preliminary data indicated that the cytoplasmic irradiation did not substantially change the active site area. In contrast, nuclear irradiation promoted an increase in the whole mitochondrial area, suggesting that nuclear DNA damage regulates the mitochondrial dynamics of the cytoplasm.

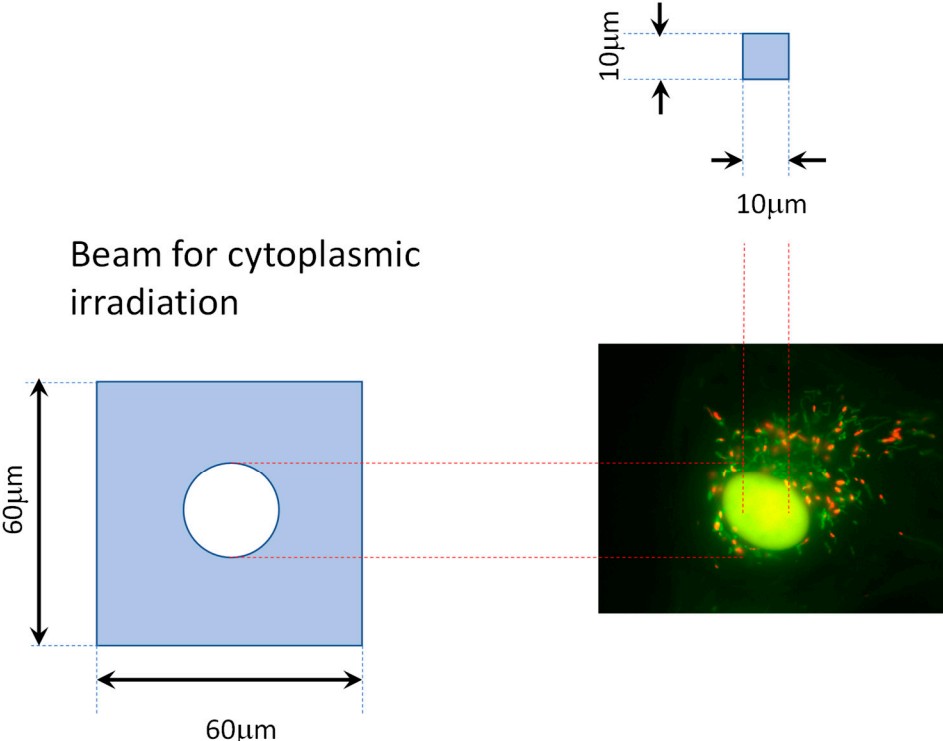

**Figure 4.** Schematic of the X-ray beam shapes for uniformly exposing cell nuclei or cytoplasm. The beam for cytoplasmic irradiation is shielded by an Au disc to avoid exposing the cell nucleus. The fine tubular structures in the photograph are mitochondria in human normal fibroblast cells (BJ-1 hTERT-Fucci) stained with mitochondrial membrane potential dye, JC-1, which emits red or green fluorescence at high or low mitochondrial membrane potential sites, respectively. The green ellipse in the cell is the cell nucleus, identified by green Fucci-fluorescence of the S/G2 phase.

## 5. Microbeam Exposure to Cell Population Systems Including 3-D Cultured Cells

Cells cultured in a dish have been used as samples for microbeam irradiation. However, the arrangement of cells in organs is different from the monolayer cells in a dish. In organs, the cells exchange signals not only through the medium, as seen with bystander signals, but also through cell-to-cell contact. We exposed a few cells in a small colony of clone cells to X-ray microbeams with a beam size of $60 \times 60~\mu m^2$ and tracked the fate of each cell by using live-cell imaging techniques [18]. HeLa-Fucci cells were used to visualize the cell cycles of the irradiated cells [19]. Typical time-lapse images of the colony are shown in Figure 5. Pedigree Tree analysis was performed for each non-irradiated cell in the colony based on the movie data (Figure 6). Cell death or prolonged cell cycle arrest of the progeny of bystander cells occurred in the colony in a dose-related fashion. Closer analysis of the pedigree Trees showed that cell death or prolonged cell cycle arrest occurred frequently in at least two of the daughter or granddaughter cells from a single parent cell. This result strongly suggests that, although the cells in the colony were clones, some of them were sensitized via the bystander effect.

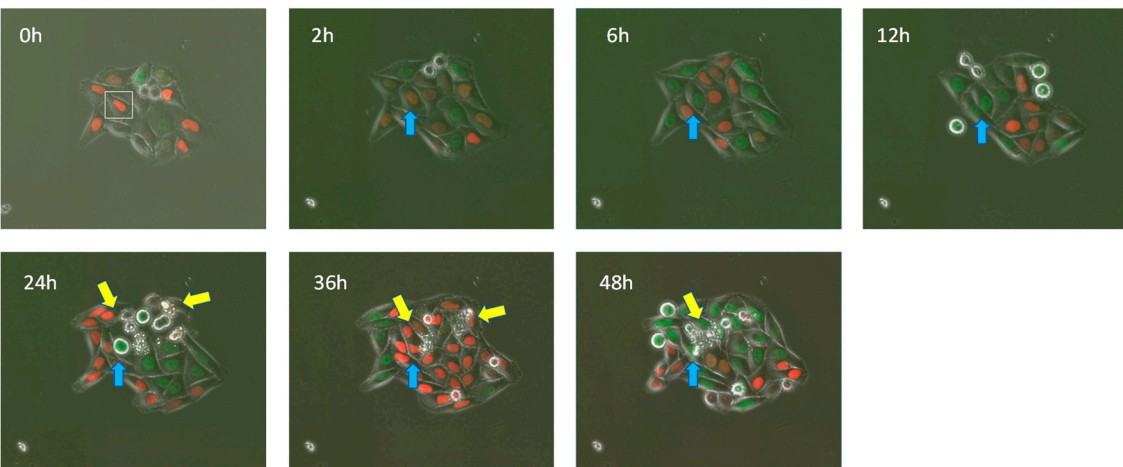

**Figure 5.** Photographs at specific times extracted from the time-lapse movie of the colony of HeLa cells. The square in the 0 h photograph and blue arrows in the others indicate the targeted cell. Yellow arrows indicate explosive cell death. This figure adapted and modified from Figure 9 in ref. [20] with permission.

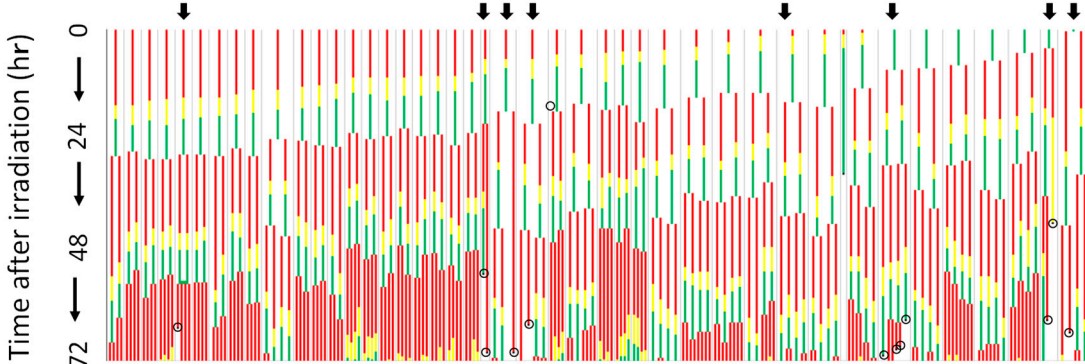

**Figure 6.** Pedigree Trees obtained from time-lapse movies of bystander HeLa-Fucci cells after several cells in the colony were exposed to 10 Gy. Red and green lines show the cell colors observed and show the G1 and G2 phases, respectively. Cells that exhibited both red and green fluorescence were assumed to be in the S phase and are shown in yellow. The trees are sorted by the duration of the first G1 or G2 phase in descending order. The circle at the end of a tree indicates explosive cell death. Arrows indicate the first cells to show cell death in their progeny. This figure was reprinted from Kaminaga et al. [18] with permission.

In addition to the monolayer cells on dishes, three-dimensionally cultured cell systems, consisting of spheroids of HeLa-Fucci cells, were exposed to X-ray microbeams with a beam size of $35 \times 35$, or $40 \times 40$ μm$^2$ [21]. The harvested spheroids were placed in a dish with a Mylar film base coated with poly(2-hydroxyethyl methacrylate) for microbeam irradiation. The spheroids were fixed to prevent movement during sample transfer from the incubator to the irradiator by filling the dish with a medium containing 0.5–1% agarose, and liquid D-MEM was added to prevent the sample from drying out. The targeting of the center of the spheroids was confirmed by cell cycle arrested cells showing green fluorescence (G2 arrested cells in Figure 7).

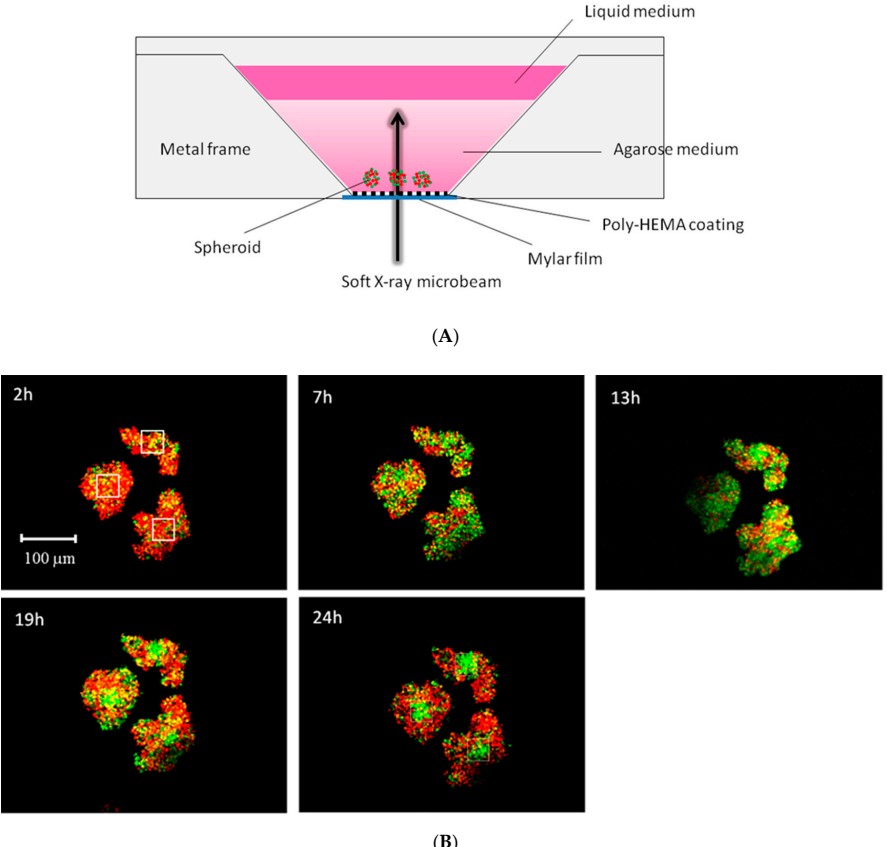

**Figure 7.** (**A**) Schematic side view of the layout of the irradiation sample of a HeLa-Fucci spheroid. (**B**) Images of the spheroids exposed to a 27 Gy X-ray microbeam. The squares in the 2 h photographs show the area exposed to a 40 × 40 μm microbeam. The images were captured at 2, 7, 13, 19, and 24 h after irradiation, as indicated in each picture. These figures were reprinted from Sakamoto et al. [21] with permission.

## 6. Clinical Appreciations of X-Ray Microbeams

Slatkin et al. [22] conducted a pioneering study of microbeam radiation therapy (MRT) by Monte Carlo simulations of dose delivery of an array of spatially fractionated synchrotron X-ray microbeams. Subsequently, many preclinical experiments using array microbeams to reduce the extent of normal cell damage have been performed, initially at the National Synchrotron Light Source at Brookhaven National Laboratory, in Upton, New York, and later at the European Synchrotron Radiation Facility (ESRF), in Grenoble, France (see review by Grotzer et al. [23] and references cited therein). Some of these studies showed that the normal central nervous system tissue in rodents and pigs can tolerate higher doses of over 100 Gy when they were delivered as spatially fractionated microbeams of several tens of micrometers [24,25]. Thus, spatially fractionated microbeams are expected to reduce undesirable adverse effects in normal tissues during radiation therapy for brain tumors. Dilmanian et al. [26] reported that certain beneficial bystander effects could lead to the tissue-sparing effect (TSE) through the release of growth factors, such as cytokines. Recently, we investigated whether the TSE could also be involved in reproductive organs regenerating germ stem cells and cell differentiation during meiosis. The combination of spatial fractionation of X-ray microbeams and a unique ex vivo testes organ culture technique [27] revealed that TSE was prominently induced in response to the non-uniform radiation fields [28]. Testes samples were obtained from *Acr-GFP* transgenic male mice around seven days postpartum (dpp) and each sample was cut into pieces approximately 1 mm$^3$ in size for the ex vivo organ culture. Spermatogenesis was monitored in the pieces of tissue by observing acrosome-green fluorescent protein (Acr-GFP) expression, which is a meiosis-specific biomarker. Each piece of tissue

was then immediately placed on a 1.5% agarose gel block immersed in α-MEM medium in a 12-well culture dish. The sample tissue on the gel block was placed in the irradiator and approximately 50% of the sample was exposed to micro-slit beams with widths of 200, 50, and 12.5 μm (Figure 8). Thus, the total energies deposited in the samples, namely, the absorbed doses, were similar (average dose: 2.5 Gy). After exposure, the GFP fluorescence from the samples was observed by fluorescent microscope for 20 days to investigate the spermatogenesis function of the sample. The testis organ samples exposed to the 200- and 50-μm micro-slit beams recovered their spermatogenesis to almost control level. However, the 12.5-μm micro-slit beam, which was comparable to the cell size (several tens of micrometers), and the wide beam that exposed the whole sample impaired spermatogenesis ([13], Figure 8). Surviving germ stem cells in the non-irradiated fields may migrate to the irradiated area and colonize it to restore the function, producing the TSE for spermatogenesis. These results indicated the distribution of the irradiation dose in the testes at the microscale level is of clinical importance for delivering high doses of radiation to tumors while preserving male fertility.

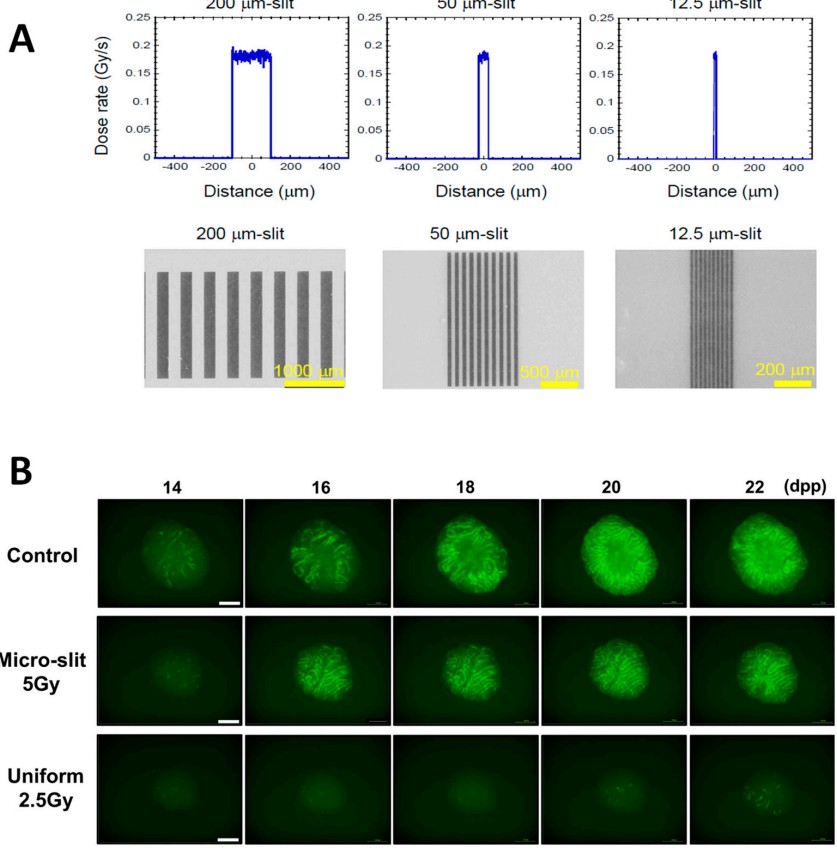

**Figure 8.** (**A**) Dose profiles of the microbeams with widths of 200, 50, and 12.5 μm, calculated with PHITS code ver. 2.96 [29] (upper three panels). The beam intensity was flat within the beamwidth. The deviation of the dose was ±6% of the averaged dose. Due to the short range of secondary electrons (1.1 μm maximum) produced by the 5.35 keV X-ray exposure, the doses delivered outside of the irradiated area were negligible (<0.25%). The dose profiles of the 200-, 50-, and 12.5-μm microbeams were experimentally confirmed using Gafchromic XR-RV3 radiochromic film (Ashland Inc., Covington, KY, USA) (lower three images). Scale bars represent 1000, 500, and 200 μm for the 200-, 50-, and 12.5-μm microbeams, respectively. These figures are reprinted from Fukunaga et al. Figure 2c,d in reference [13]. (**B**) Representative images show *Acr-GFP* expression changes in single cultures following a dose of 5 Gy from a 200-μm-wide microbeam, 2.5 Gy from uniform X-ray irradiation, and 0 Gy (control), from 12 to 20 dpp. The photographs at 8 dpp are bright-field images. These figures are reprinted from Fukunaga et al. Figure 3a,b in rereference [13].

## 7. Summary

Synchrotron radiation from the Photon Factory is a source of quasi-parallel X-ray microbeams that can be used as a powerful probe to target specific sites in a living system. Using the beam, mechanistic studies of bystander responses of the cells, as well as the effects on organelles, such as mitochondria in the cytoplasm, have been studied extensively over the last decade. Spatial beam fractionation (micro-slit beams) is a promising technique that could be used in clinical radiation therapy of tumors near the testes.

**Author Contributions:** Conceptualization, A.Y. and N.U.; writing—Original draft preparation, A.Y.; writing—Review and editing, N.U. All authors have read and agreed to the published version of the manuscript.

**Funding:** This research received no external funding.

**Acknowledgments:** We are grateful to our colleagues, H. Fukunaga and K. Kaminaga, who performed our experimental studies involving synchrotron microbeam experiments. The authors thank K. Fujii and M. Noguchi of the National Institutes of Quantum and Radiological Sciences for their helpful assistance with cell experiments and beamtime. The work using synchrotron X-ray microbeams described in this paper was approved by the Photon Factory Program Advisory Committee (Proposal No. 2013G214, 2015T001, and 2017G565) and supported by a Japan Society for the Promotion of Science KAKENHI grant (No. 15H02823).

**Conflicts of Interest:** The authors report no declaration of interest. The authors alone are responsible for the content and writing of the paper.

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
