# Peer review of "Targeting Specific Sites in Biological Systems with Synchrotron X-Ray Microbeams for Radiobiological Studies at the Photon Factory"

_qubs, doi:10.3390/qubs4010002_

Round 1
Reviewer 1 Report
The paper is interesting. The authors review investigations on radiation-induced non-targeted effects induced by targeted exposure of specific sites to synchrotron X-ray microbeams mainly from the KEK IMSS Photon Factory microbeam. The paper first presents the different ways to focus the synchrotron X-ray beam at subcellular level (cell nucleus or cytoplasm) thanks to Fresnel zone plates or to simple metal slit system (gold masks). Then, authors mention microbeam experiments from cytoplasmic and mitochondria irradiation on cell culture monolayers to 3D cultures cells with spheroid Hela-Fucci cells. At the end, they introduce also micro-slit beams for spatial beam fractionation as promising technique.
The paper is well written and the title and the abstract well describe its content.
Major concerns
Line 133: Unpublished data are mentioned as reference. I would not refer to unpublished data. If accepted by editors, more details are needed.
Minor concerns
In general, when data is available, it would be good to add beam size at the level of cell for each experiment.
Line 41-42: references are missing
Line 34: Correct the reference Hall and Hei instead of Holl and Hei.
Line 42 to 45: The transition sentence need to be modified: “Early studies highlighted bystander effect but did not elucidate the whole underlying mechanisms of the bystander effect because. Indeed, as the alpha particles hit the cells in a stochastically random manner, the cells that were hit could not be identified spatially and temporally in the culture dish.”
Line 46-47: References are needed. All experiments done on particle microbeam dealing with non-targeted effect and mainly to bystander effect are not taking into account. As you mention previously papers using alpha particle broadbeam to introduce bystander effect, experiments on the subject with alpha particle microbeam needs to be mentioned too as references at least.
Cytoplasmic irradiation with X-ray microbeams
Line 133: if unpublished results are accepted by editors, it would be better to give more information.Author Response
Authors’ answers to Reviewer #1
Thank you very much for your useful comments and advice. Our point-by-point answers to your comments are below. The revised text is shown in blue in the manuscript.
Major concerns
Line 133: Unpublished data are mentioned as reference. I would not refer to unpublished data. If accepted by editors, more details are needed.
We apologize for including a reference to unpublished data. Because the paper has not been accepted yet, we have revised this part as follows.
“The particular mitochondrial active sites with high membrane potential were visualized by fluorescent chemical probes. Unexpectedly, preliminary data indicated that the cytoplasmic irradiation did not substantially change the active site area. In contrast, nuclear irradiation promoted an increase in the whole mitochondrial area, suggesting that nuclear DNA damage regulates the mitochondrial dynamics of the cytoplasm.”
Minor concerns
In general, when data is available, it would be good to add beam size at the level of cell for each experiment.
We have added the beam sizes for each experimental description where they were missing in the original manuscript. The additional beam sizes are as follows. Line numbers refer to the original manuscript.
Line 75, “using a simple metal slit system (beam size from 5 μm to several millimeters)”
Line 78, “human fibroblast cells (WI-38) were targeted with 10 × 10 μm2 beam”
Line 128, “Cytoplasmic irradiation was performed using a 60 × 60 μm2”
Line 146, “clone cells to X-ray microbeams with a beam size of 60 × 60 μm2”
Line 168, “HeLa-Fucci cells, were exposed to X-ray microbeams with a beam size of 35 × 35 or 40 × 40 μm2”
Line 41-42: references are missing
We have added a reference for the bystander signal transduction and genetic instability by as reference (6): Lorimore SA, Wright EG. Radiation-induced genomic instability and bystander effects: related inflammatory-type responses to radiation-induced stress and injury? A review. Int J Radiat Biol. 2003;79(1):15-25.
Line 34: Correct the reference Hall and Hei instead of Holl and Hei.
This has been corrected.
Line 42 to 45: The transition sentence need to be modified: “Early studies highlighted bystander effect but did not elucidate the whole underlying mechanisms of the bystander effect because. Indeed, as the alpha particles hit the cells in a stochastically random manner, the cells that were hit could not be identified spatially and temporally in the culture dish.”
We have corrected this part in accordance with your suggestion as follows.
“Early studies observed the bystander effect but did not elucidate the entirety of the underlying mechanisms of the bystander effect. Because the alpha particles hit the cells in a stochastically random manner, the cells that were hit could not be identified spatially and temporally in the culture dish.”
Line 46-47: References are needed. All experiments done on particle microbeam dealing with non-targeted effect and mainly to bystander effect are not taking into account. As you mention previously papers using alpha particle broadbeam to introduce bystander effect, experiments on the subject with alpha particle microbeam needs to be mentioned too as references at least.
We have added three review papers as references for alpha particles (He2+) or other charged particle microbeams from accelerators. The description added to the last paragraph of the introduction is as follows.
“Various high-LET charged particle microbeams of protons, He, carbon, or much heavier ions, such as uranium, have been used to study the correlation between an extremely localized track structure and the resulting complex DNA damage induction followed by enzymatic repair in cultured cells (see reviews by Tobias et al. (7), Barberet and Seznec (8), and Kobayashi et al. (9)).”
Cytoplasmic irradiation with X-ray microbeams
Line 133: if unpublished results are accepted by editors, it would be better to give more information.
We have substantially changed this part (see above).

Reviewer 2 Report
This paper presents a “review” of radiobiological works done using synchrotron radiation generated X-ray microbeams. The paper is generally well written and easy to read and follow.
Despite its readability the manuscript presents a few fundamental flaws.
The paper reviews works done by a (few) team(s) at a single facility (Photon Factory, Tsukuba). However, several groups at different facilities located around the world are dealing with microbeam-induced bystander effects. An excellent (but absolutely not exhaustive!) reference is the review by Drexler GA, Ruiz-Gómez MJ. Microirradiation techniques in radiobiological research, J Biosci. 2015 Sep;40(3):629-43, which reviewed the state of the art. I see here two different possibilities: a) authors could decide to update the Drexler’s review, transforming their report in a real review paper; b) author can continue along the same trace of this manuscript, provided that they very clearly state the limits of their work, in particular in the title and in the introduction, by clearly indicating that the review is limited to work done at the PF. In all cases, a more extensive reference to recent works by colleagues around the world is requested, to better situate the author’s work with respect to the state of the art. I have a real problem in letting include unpublished and therefore not fully peer reviewed data in a “review”. There are no ways to understand whether the mentioned work is free from flaws (missing all background, the details on methodology and the discussion information typical of any peer-reviewed paper); the inclusion of these unpublished results in this “review” will legitimate a not reviewed work in a reviewed paper. Following this logic, this part must be removed from the text. 217-219 Claims in the summary needs to be significantly lowered. The cited paper, in an ex-vivo environment, only shows one potential benefit of microbeams vs full beam. The way to clinics has to include full preclinical studies on tissue sparing and on therapeutic effects, using radiotherapy compatible energies: all of this has yet to be started.
Other important points to be addressed:
These studies challenged the radiobiological dogma that radiation damage to DNA is the major cause of genomic alterationI would rather lower the concept by adding the word “only” before “major”, unless author have more refined proofs
83, 104-107,177 10 × 10 μm
Add square to microns
Indicate the source of Figure 5 in the caption
Missing capitalized letter on “Trees”
Synchrotron radiation is a quasi parallel source, not perfectly parallel. Also, optics (like the here indicated Fresnel lenses) modifies the beam divergence. So the concept must be modified
A section on the problematics regarding the dose measurement and definition would be essential. The concept of dose, in these small volumes, is something that needs to be carefully reviewed, in order to be able to compare, in a meaningful way, the different results.
Author Response
Authors’ answers to Reviewer #2
Thank you for your sound advice and useful comments. We agree with your comments. In accordance with your suggestion (b), namely, “author can continue along the same trace of this manuscript, provided that they very clearly state the limits of their work, in particular in the title and in the introduction, by clearly indicating that the review is limited to work done at the PF”, we have clarified that the manuscript is intended to refer to research activities conducted at the Photon Factory through the following changes to the manuscript. The revised text is shown in green in the manuscript.
The paper title was changed to clarify the fact that the microbeam studies were conducted at the Photon Factory: “Targeting specific sites in biological systems with synchrotron X-ray microbeams for radiobiological studies at the Photon Factory”. We changed the first sentence of the introduction as follows.
“Introduction
Cells in multicellular systems have been targeted by irradiation with ionizing radiation or UV lasers in mechanistic studies of radiobiological phenomena. These phenomena ultimately lead to carcinogenesis induced by low-dose radiation from the environment or medical applications, such as microsurgery or microbeam radiation therapy (MRT) (see review by Drexler and Ruiz-Gómez (1)).”
To specify the target of this paper, we added the following text to the last paragraph of the introduction.“On the other hand, the synchrotron X-ray microbeam provides a uniformly exposed small area in a wide range of biological targets, from DNA to animals. This is a specific advantage of this microbeam compared with ion particle irradiation or focused laser beams. In this review, recent advances in mechanistic studies of specific sites in biological systems targeted with a parallel X-ray microbeam from a synchrotron facility, the Photon Factory, in Tusukuba, Japan, are described.”
We apologize for referencing unpublished data. We have changed this part as follows by “Blue” to respond to another reviewer who also gave us a similar suggestion.“The particular mitochondrial active sites with high membrane potential were visualized by fluorescent chemical probes. Unexpectedly, preliminary data indicated that the cytoplasmic irradiation did not substantially change the active site area. In contrast, nuclear irradiation promoted an increase in the whole mitochondrial area, suggesting that nuclear DNA damage regulates the mitochondrial dynamics of the cytoplasm.”
To cite more references about clinical applications reported by other groups, we have added the subsection “Clinical applications of X-ray microbeams” to provide a substantial description of these studies.
“Clinical applications of X-ray microbeams
Slatkin et al. (15) conducted a pioneering study of microbeam radiation therapy (MRT) by Monte Carlo simulations of dose delivery of an array of spatially fractionated synchrotron X-ray microbeams. Subsequently, many preclinical experiments using array microbeams to reduce the extent of normal cell damage have been performed, initially at the National Synchrotron Light Source at Brookhaven National Laboratory, in Upton, New York, and later at the European Synchrotron Radiation Facility (ESRF), in Grenoble, France (see review by Grotzer et al. (16) and references cited therein). Some of these studies showed that the normal central nervous system tissue in rodents and pigs can tolerate higher doses of over 100 Gy when they were delivered as spatially fractionated microbeams of several tens of nanometers (17, 18). Thus, spatially fractionated microbeams are expected to reduce undesirable adverse effects in normal tissues during radiation therapy for brain tumors. Dilmanian et al. (19) reported that certain beneficial bystander effects could lead to the tissue-sparing effect (TSE) through the release of growth factors, such as cytokines. Recently, we investigated whether the TSE could also be involved in reproductive organs regenerating germ stem cells and the cell differentiation during meiosis.”
Other important points to be addressed:
These studies challenged the radiobiological dogma that radiation damage to DNA is the major cause of genomic alteration. I would rather lower the concept by adding the word “only” before “major”, unless author have more refined proofs.We have changed this to “dogma that radiation damage to DNA is the only major cause of genomic alteration” here and throughout the text.
83, 104-107, 177 10 × 10 μm Add square to microns.
This has been corrected accordingly.
Indicate the source of Figure 5 in the caption.
The source of this figure was added to the caption.
“This figure adapted and modified from Figure 9 in ref (28) with permission.”
Missing capitalized letter on “Trees”
This has been corrected accordingly.
Synchrotron radiation is a quasi parallel source, not perfectly parallel. Also, optics (like the here indicated Fresnel lenses) modifies the beam divergence. So the concept must be modified.
We have replaced “parallel source” with “quasi-parallel source”. We added the following sentence about the Fresnel zone plates to the first paragraph of the section “Synchrotron X-ray microbeams in radiation biology”.
“This is an advantage of the simple optical setup compared with a focusing system based on an X-ray Fresnel zone plate, which gives a large beam divergence even when it is inserted into a synchrotron beamline.”
A section on the problematics regarding the dose measurement and definition would be essential. The concept of dose, in these small volumes, is something that needs to be carefully reviewed, in order to be able to compare, in a meaningful way, the different results.To address the concept of dose, we added the following sentences to the end of the first paragraph in the section “Synchrotron X-ray microbeams in radiation biology”.
“The method of evaluating the absorbed dose in the exposed area should also be discussed. Due to the short range of secondary electrons (1.1 μm maximum) produced by 5.35 keV X-ray exposure, the dose delivered outside of the irradiated area was negligible. Fukunaga et al. (9) used PHITS code to calculate dose profiles of various spatially fractionated microbeams with widths of 12.5, 50, and 200 μm for ex vivo testes organ culture (details are described in the section “Clinical applications of X-ray microbeams”). The estimated dose outside of the exposed area was less than 0.25% of that of the exposed one.”

Reviewer 3 Report
This is an interesting paper shedding light on debated topics such as the bystander effect and providing a novel hypothesis involving mithochondrial and cytoplasmic damage as an additional factor causing cell damage and death.
Author Response
Authors answers to Reviewer #3
We thank the reviewer for reading our paper and recognizing its importance in radiobiology research.
Round 2
Reviewer 2 Report
Authors have satisfactorily addressed all my previous concerns. However, the added text contains some issues that need to be corrected before publication:
Clinical appreciations of X-ray microbeams: the size of the beams reported in the two cited papers are not "several tens of nanometers" as indicated, but several tens of micrometers. Clinical appreciations of X-ray microbeams: missing a (capitalized) letter after (23,24)Author Response
Authors’ answers to Reviewer #2
Clinical appreciations of X-ray microbeams: the size of the beams reported in the two cited papers are not "several tens of nanometers" as indicated, but several tens of micrometers. Clinical appreciations of X-ray microbeams: missing a (capitalized) letter after (23,24)
These have been corrected accordingly.